# AI-designed NMR spectroscopy RF pulses for fast acquisition at high and ultra-high magnetic fields

V. S. Manu [1], Cristina Olivieri [1,2] & Gianluigi Veglia [1] ✉

Nuclear magnetic resonance (NMR) spectroscopy is a powerful high-resolution tool for characterizing biomacromolecular structure, dynamics, and interactions. However, the lengthy longitudinal relaxation of the nuclear spins significantly extends the total experimental time, especially at high and ultra-high magnetic field strengths. Although longitudinal relaxation-enhanced techniques have sped up data acquisition, their application has been limited by the chemical shift dispersion. Here we combined an evolutionary algorithm and artificial intelligence to design [1]H and [15]N radio frequency (RF) pulses with variable phase and amplitude that cover significantly broader bandwidths and allow for rapid data acquisition. We re-engineered the basic transverse relaxation optimized spectroscopy experiment and showed that the RF shapes enhance the spectral sensitivity of well-folded proteins up to 180 kDa molecular weight. These RF shapes can be tailored to re-design triple-resonance experiments for accelerating NMR spectroscopy of biomacromolecules at high fields.

Nuclear magnetic resonance (NMR) spectroscopy plays an essential role in structural biology, providing atomic-resolution information about the structure, motions, and interactions of biological macromolecules in near physiological conditions[1]. Originally, NMR spectroscopy was relegated to the analysis of small and medium size biomacromolecules. However, the development of transverse relaxation optimized spectroscopy (TROSY)[2] gave a new thrust to NMR, enabling the characterization of biological assemblies up to 1 MDa molecular weight[3]. Today, the combination of advanced NMR techniques with high and ultra-high magnetic fields as well as deuteration and methyl group isotopic labeling makes it possible to study even larger complexes[4–6].

However, as the strength of magnetic fields increases, the design of RF pulses and pulse sequences must account for more intense radiation damping, broader chemical shift dispersion, chemical exchange, and field inhomogeneity, phenomena that are accentuated at higher frequencies[7]. Additionally, NMR transitions are characterized by relatively long relaxation times[8]. Although long transverse spin relaxation times are desirable as they result in sharper lines, lengthy longitudinal spin relaxation times can be detrimental as they require inter-scan delays on the order of seconds to re-establish the spin magnetization equilibrium, extending the total experimental time, especially for large biomacromolecules at high and ultra-high magnetic fields. As a result, all NMR spectrometers spend most of the time idle, waiting for the spin systems to return to equilibrium after a pulse or a pulse sequence. A solution to this problem is to use longitudinal relaxation-enhanced (LRE) techniques[9]. One approach involves doping the samples with relaxation-enhanced agents, such as paramagnetic ions, that accelerate the longitudinal relaxation of the spin systems, though these agents may broaden the NMR lines and, in some cases, affect the sample's chemistry[10]. An alternate approach is to use polarization-enhanced fast-pulsing techniques[9,11] such as Band-selective Excitation Short-Transient (BEST)[12], SOFAST[13], and ULTRA-FAST[14,15] methods. Since the relaxation of a proton spin in a molecular system depends on the spin states of nearby protons and bulk water, these techniques utilize band-selective excitation pulses on an isolated group of nuclear spins (e.g., amides resonating at a lower field relative to the aliphatic protons), enabling their faster recovery to a thermal

[1]Department of Biochemistry, Molecular Biology & Biophysics and Department of Chemistry, University of Minnesota, Minneapolis, MN 55455, USA. [2]Present address: Department of Chemistry, University of Milan, 20133 Milan, Italy. ✉e-mail: vegli001@umn.edu

equilibrium[9]. As a result, the inter-scan delays can be significantly shortened from seconds (required for relaxation of the bulk water) to hundreds of milliseconds (for relaxation of amide resonances), speeding up data acquisition.

The most common way to achieve selective excitation of amide protons is to replace standard hard pulses in the heteronuclear single quantum coherence (HSQC) or heteronuclear multiple quantum coherence (HMQC) pulse sequences with band-selective pulses[16] for excitation and refocusing operation of the amide protons[11]. For instance, the BEST-type experiments utilize band-selective pulses such as PC9[17], EBURP-2[18], REBURP[18], and r-SNOB[19,20]. These pulses have been implemented in the BEST-TROSY multidimensional NMR experiments, significantly shortening the total experimental time[12], and found application specifically for intrinsically disordered proteins (IDPs)[21]. However, the application of BEST methods to folded proteins with well-dispersed amide resonances remains sparse. The latter is due to the limitations imposed by the band-selective pulses that achieve partial excitation of resonances at the fringes of the irradiation bandwidth, which in turn causes a significant loss of sensitivity for amide protons resonating near the water signal or de-shielded by ring-current effects. These issues are even more apparent for well-folded proteins with significant β-sheet contents or carrying paramagnetic centers and are certainly accentuated at high and ultra-high magnetic fields. An obvious solution to this problem is reducing the selectivity of the band-selective pulses. However, this expedient leads to partial excitation of the water signal that, together with off-resonance effects, may generate a loss of steady-state polarization and a net reduction of the overall sensitivity[9].

Here, we utilized GENETICS-AI, or GENErator of TrIply Compensated RF pulSes via Artificial Intelligence[22], to design fast-pulsing experiments that address the technical challenges posed by ultra-high magnetic field spectroscopy. GENETICS-AI combines an evolutionary algorithm and artificial intelligence (AI) to generate inhomogeneity-compensated high-fidelity RF pulses[22]. The original version of GENETICS-AI searched for optimal phase space to create RF shapes at a constant amplitude[22,23]. The current version presented here simultaneously explores RF phase and amplitude space, generating highly compensated pulses (i.e., [1]H broadband universal π and band-selective pulses) devoid of water and aliphatic protons irradiation that covers a significantly broader bandwidth relative to the commonly used band-selective pulses. Additionally, these RF shapes are time optimal, enabling both chemical shift and J coupling evolution during their execution. Using these RF pulses, we re-engineered the classical TROSY-HSQC experiment into a RAPID (RApid PulsIng broaDband) TROSY experiment and tested its performance with three proteins: the 21 kDa Raf kinase inhibitor protein (RKIP), 42 kDa maltose binding protein (MBP), and the 180 kDa dimer of the isoform IIβ of the regulatory subunit of protein kinase A (RIIβ). The RAPID-TROSY experiment outperforms the previous hetero-correlated experiments for sensitivity, homogeneity compensation, speed of data acquisition, and operational bandwidth; therefore, it constitutes a template for accelerating NMR spectroscopy of large biomacromolecules at high and ultra-high magnetic fields.

## Results

### AI-designed phase and amplitude-modulated RF pulses

Using GENETICS-AI, we generated four universal spin operations that are common to all 2D and 3D experiments for biomolecular NMR spectroscopy: (i) a 90° pulse for band-selective excitation (UA90ev1) with chemical shift and J evolution, (ii) a 180° pulse for [1]H refocusing with J evolution (URev1), (iii) a 180° pulse for [15]N refocusing with J evolution (URev2), and (iv) a 180° band-selective pulse centered on the amide protons that avoids water irradiation (UARev1). Figure 1 illustrates the RF pulses' amplitude and phase *vs.* time and the simulated offset responses for the magnetization components ($M_x$, $M_y$, and $M_z$). The duration,

amplitude, and operational bandwidth of these pulses are detailed in Supplementary Table 1, and their 2D profiles of the amplitude-offset responses for the magnetization components are depicted in Supplementary Figs. 1 and 2. These plots show that the RF pulses cover a significantly broader bandwidth relative to the band-selective pulses commonly used in the BEST type of experiments. As a comparison, we report the response profiles for the PC9-90 and REBURP pulse with typical bandwidths ranging from 0 to 4 kHz (Supplementary Fig. 3).

To quantify the extent of J coupling evolution during the execution of the RF pulses, we calculated the magnitude of the antiphase operators ($2I_xS_z$ and $2I_yS_z$) as a function of the offset (Fig. 1). Specifically, we simulated three different schemes: a) pulsing on one channel (blue traces), b) pulsing on both channels (red traces), and c) no pulsing, i.e., including a delay equivalent to the pulse duration (green traces). The UA90ev1 pulse selectively excites the proton resonances in the 0.5–7.5 kHz range while leaving the water (on resonance) and other protons resonating between 0 and −6.2 kHz unaffected. Note that both chemical shifts and J coupling evolution occur between 0.5 and 7.5 kHz (Fig. 1A). The URev1 pulse is designed to refocus chemical shifts and allow for J coupling evolution during pulsing on both channels (Fig. 1B). The URev2 pulse performs a refocusing operation on the [15]N channel over the entire operational bandwidth (Fig. 1C). Finally, the UARev1 pulse avoids water irradiation and refocuses the resonances of interest in the range of 0.5 to 7.5 kHz, enabling chemical shift and J coupling evolution (Fig. 1D). Since the duration of the GENETICS-AI excitation and refocusing pulses is longer than the typical hard pulses (Supplementary Table 1), we also assessed the effects of relaxation during their execution. Specifically, we performed numerical simulations and quantified the magnitude of the module of the magnetization vector ($|M_{xyz}| = \sqrt{M_x^2 + M_y^2 + M_z^2}$) as a function of the longitudinal ($R_1$) and transverse ($R_2$) relaxation rates (Supplementary Figs. 4 and 5). These simulations were carried out with different initial states of the magnetization components ($M_x$, $M_y$, and $M_z$) and a pulse duration of 1 ms. For the excitation pulses, we compared the effects of relaxation of the PC9_90 pulse, a typical shape used for [[1]H,[15]N] BEST-TROSY experiments, with the UA90ev1 pulse. As a reference, we simulated the case in which no RF pulsing, i.e., including a 1 ms delay corresponding to the pulse duration. When no pulse is applied (Supplementary Fig. 4A), the magnitude of the $M_x$ and $M_y$ components decreases with increasing the transverse relaxation time, whereas the longitudinal relaxation has no apparent effects within the range of 0–50 s$^{-1}$. In contrast, the $M_z$ component remains essentially unaffected. For the PC9_90 pulse, the transverse relaxation significantly affects both the $M_x$ and $M_y$ components, with a notable reduction of the $M_z$ component. For the UA90ev1 pulse (Supplementary Fig. 4C), the effects of the transverse relaxation on the $M_x$ and $M_y$ components are remarkably similar to the no pulsing case, and the effects on the $M_z$ component are negligible. We repeated the calculations for all refocusing pulses using the REBURP pulse as a reference (Supplementary Fig. 5). As previously inferred[18], we found that transverse relaxation significantly impacts the performance of the REBURP pulse (Supplementary Fig. 5A), especially at faster relaxation rates. The URev1 pulse shows a similar response for $M_y$ and $M_z$ components, whereas the $M_x$ component is slightly more affected by both $R_1$ and $R_2$. The URev2 pulse responded to the relaxation mechanisms similar to the REBURP pulse for all three magnetization components. Nonetheless, both URev1 and URev2 have a significantly broader operational bandwidth and more homogenous irradiation for the amide region than the REBURP pulse. Finally, the band-selective UARev1 pulse displays a similar behavior for $M_x$ and $M_y$, though the $M_z$ component remains essentially unaffected. Taken together, these simulations show that the performance of our RF pulses is similar to or better than the canonical band-selective pulses with minimal relaxation effects, making them suitable for multidimensional pulse sequences.

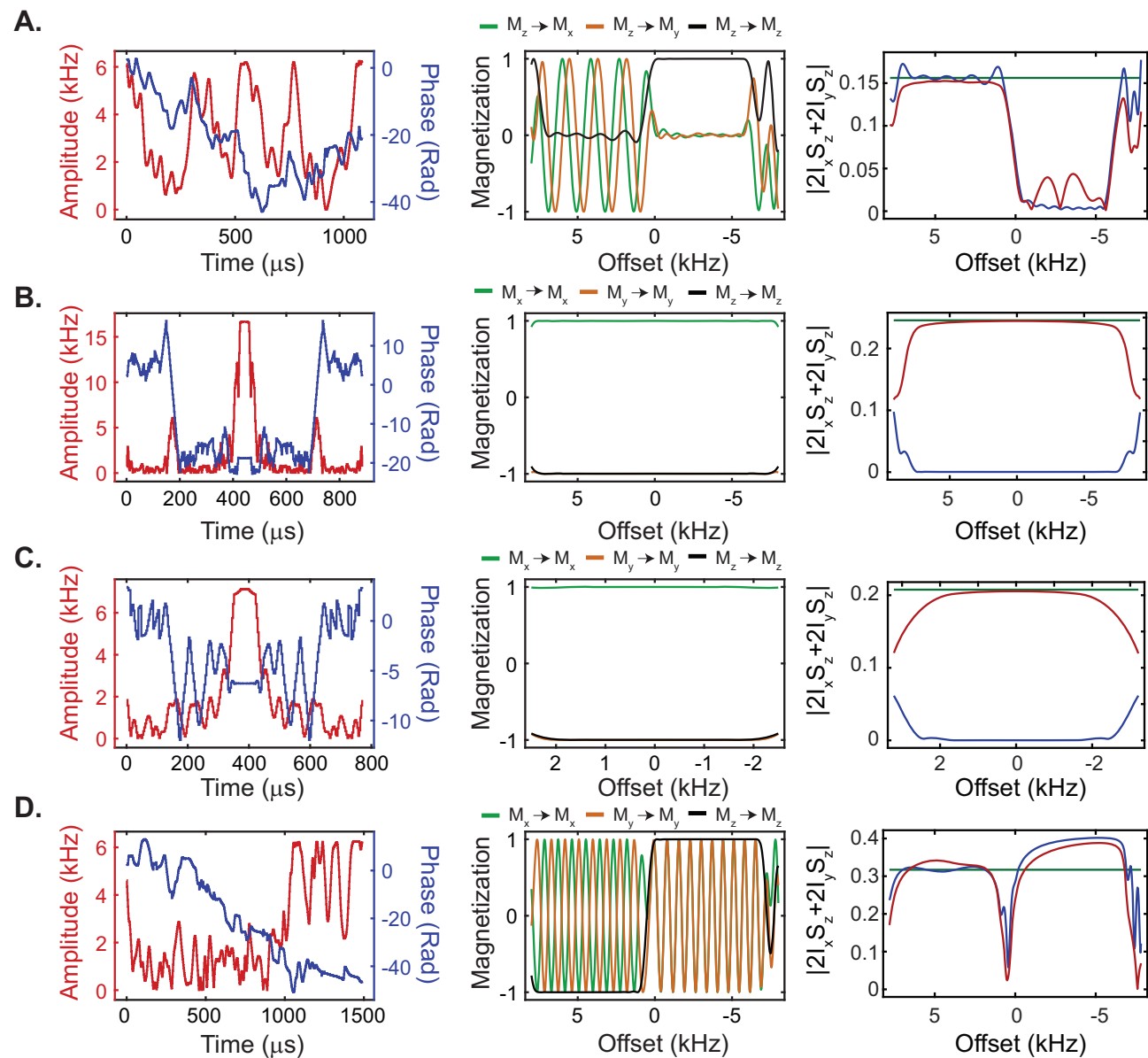

**Fig. 1 | RF pulse shapes designed using the GENETICS-AI software.** RF pulse amplitudes and shapes (left), offset responses (center), and J coupling evolution profiles (right) for **A**. the band-selective 90° pulse UA90ev1, **B**. the broadband 180° pulse URev1, **C**. the broadband 180° pulse URev2, and **D**. the band-selective 180° UARev1. The effects of the pulses on the magnetization components are color-coded in the graph with the offset response. The oscillations of the $M_x$ and $M_y$ components of the magnetization indicate the chemical shift evolution during the pulse. The J coupling evolution was calculated using the two spin operators, $2I_xS_z$ and $2I_yS_z$, during the pulse duration, with the UA90ev1 pulse starting from an initial magnetization $\rho_{initial}$: $I_z$, the URev1 from $\rho_{initial}$: $I_x$, the URev2 from $\rho_{initial}$: $I_x$, and the UARev1 pulse from $\rho_{initial}$: $I_x$. Note that pulsing on a single channel refocuses the J coupling evolution, whereas pulsing on both channels restores the J coupling evolution. The green traces indicate the offset response for ideal instantaneous pulses. The blue traces are the offset response for pulsing on a single channel (I or S), and the red traces are the offset response for pulsing on both channels (I and S). The duration and operational bandwidth of the pulses are reported in Supplementary Table 1, and the 2D response profiles are shown in Supplementary Fig. 1.

## Re-engineering the [¹H,¹⁵N] TROSY-HSQC pulse sequence with GENETICS-AI RF shapes

Using the GENETICS-AI RF shapes, we re-engineered the canonical [¹H,¹⁵N] TROSY-HSQC experiment[2] (hereinafter referred to as [¹H,¹⁵N] TROSY), which is the building block for multidimensional NMR experiments for large biomacromolecules (Fig. 2)[24]. We first modified the INEPT scheme, implementing a UA90ev1 to flip the amide protons onto the transverse plane while leaving the aliphatic and water resonances unperturbed along the $z$-axis (Fig. 1A). During this operation, the chemical shifts and J couplings evolve for the entire duration of the pulse ($T_{p1}$), i.e., ($T_{p1}/2$)–U(90)–($T_{p1}/2$), where U(90) is the RF operator for the 90° pulse ($e^{-i(\pi/2 I_x)}$). The 90° pulse acts as a universal

flipping operation, converting the z magnetization into -y. During the subsequent delay the magnetization dephases linearly with the pulse length and chemical shift offset (Fig. 1A). Since the 90° pulse and delays are part of the entire spin echo sequence, there is no net effect on the phase of the magnetization.

We then replaced the two hard refocusing pulses on the ¹H and ¹⁵N channels with URev1 and URev2, respectively. During these pulses, both chemical shift and J coupling operators evolve for 98% of the time ($T_{p3}$), i.e., ($0.49 \times T_{p3}$) – U(180) – ($0.49 \times T_{p3}$). Since the 180° operation is sandwiched between two delays, the dephasing of the magnetization caused by the first delay is refocused by the second delay. When applied simultaneously with a 180° pulse on the second channel, these

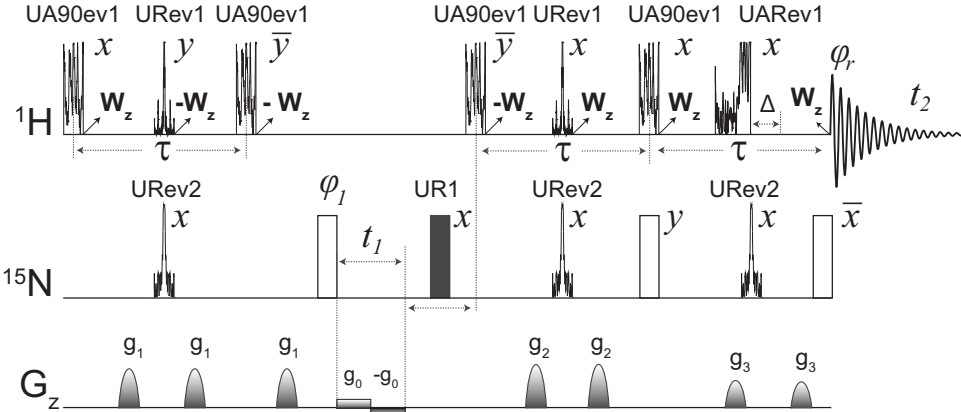

**Fig. 2 | Pulse scheme for the [¹H,¹⁵N] RAPID-TROSY experiment.** The pulses generated by the GENETICS-AI software for ¹H and ¹⁵N channels are reported according to the optimized amplitude shapes generated. Open rectangular pulses in the ¹⁵N channel are 90° hard pulses and were not substituted in the original sequence. The pulse operations for the GENETICS-AI pulses are summarized in Table S1 and explained in the main text. The delay, $\Delta$, associated with UARev1 is 0.75 times the length of the UARev1 pulse. The delay $\tau$ is equal to $1/(2J_{HN})$, where $J_{HN}$ is the coupling constant between ¹H and ¹⁵N. The phase cycle $\varphi_1 = (y, -y, -x, x)$ for odd and $\varphi_1 = (-y, y, -x, x)$ for even $t_1$ points. Receiver phase cycle $\varphi_r = (x, -x, -y, y)$. The ¹H magnetization of water (W) and aliphatic protons is kept along the $z$-axis throughout the pulse sequence. The amplitude of the gradient pulses, $g_1, g_2$, and $g_3$, applied along the $z$-axis were 1.97, 2.96, and 3.29 Gauss.

delays cause the magnetization to evolve under the J coupling. This evolution can be exploited during spin echo sequences for polarization transfer, reducing the length of the spin echo. In both cases, however, there is no net evolution of the chemical shift. We then used a second UA90rev1 pulse with phase -y and a ¹⁵N 90° pulse with phase $\varphi_1$ to transfer the polarization from ¹H to the ¹⁵N nuclei enabling chemical shift evolution during $t_1$.

Next, we applied a phase-modulated broadband pulse at constant amplitude (UR1), which we previously designed[23], to refocus any unwanted ¹⁵N chemical shift evolution and eliminate phase errors. We then mirrored the above scheme for the sensitivity-enhanced reverse-INEPT sequence that enables to record ¹H coherences (inverse detection). To selectively detect the amide protons and eliminate the water signal, we utilized the UARev1 pulse, which avoids water and aliphatic protons irradiation, leaving their magnetization along the z-axis. During UARev1, the amides' chemical shift and J coupling operators evolve for 75% of the time ($T_{p2}$), i.e., $(0.75 \times T_{p2})$ – U(180), where U(180) is the operator for the 180° pulse $\left(e^{-i(\pi I_x)}\right)$, and the symmetric spin-echo operation is achieved including a delay, $\Delta$, equivalent to 0.75 times the pulse duration. This delay ensures that there is no effect on the magnetization phase. The UARev1 pulse and $\Delta$ are centered relative to the refocusing pulse applied on the ¹⁵N channel (URev2), and its response to $\Delta$ is shown in Supplementary Fig. 2. The slow relaxing component of the ¹H-¹⁵N multiplet for the TROSY effect is selected via phase cycling[2]. Note that the URev1 pulse on the ¹H channel is not band-selective; hence, the inversion operation of the water and aliphatic proton magnetization during the INEPT transfer is compensated by an equivalent operation in the reverse-INEPT sequence.

## Applications of [¹H,¹⁵N] RAPID-TROSY to globular proteins and comparison with standard TROSY schemes

To test the [¹H,¹⁵N] RAPID-TROSY pulse sequence, we prepared a sample of uniformly (U) ¹⁵N labeled RKIP, a small globular protein of 21 kDa, and compared its amide fingerprint spectrum with the corresponding spectra obtained using [¹H,¹⁵N] BEST-TROSY[21] and [¹H,¹⁵N] TROSY-HSQC[25,26] experiments (*IBS_BTROSY* and *trosyetf3gpsi2* sequences in the Bruker library). We systematically varied the interscan delay (D1) for the three pulse sequences and analyzed the relative sensitivity gains (Fig. 3). For comparison, we included our previously published [¹H,¹⁵N] WADE-TROSY experiment[23] to establish whether the

signal intensity increase is due to the water suppression scheme. In the [¹H,¹⁵N] WADE-TROSY experiment, the conventional water suppression scheme is substituted by a water irradiation devoid (WADE) pulse[23]. WADE pulses are designed with 'π-shifted' symmetry, creating a null point at the water resonance. Compared to conventional water suppression methods, WADE pulses exhibit greater sensitivity and offer adjustable water selectivity to prevent the suppression of amide resonances near the water signal. As a result, the WADE pulses confer higher sensitivity, especially for exchanging amide sites.

We found that at short D1 values (0.2−0.25 s), the [¹H,¹⁵N] BEST-TROSY and [¹H,¹⁵N] RAPID-TROSY experiments are the most sensitive (Fig. 3A). As D1 increases, the performance of the pulse sequences diverges, with the [¹H,¹⁵N] WADE- and RAPID-TROSY experiments showing a monotonic increase in sensitivity gain. Notably, at D1 longer than 1.0 s, the [¹H,¹⁵N] WADE-TROSY and [¹H,¹⁵N] RAPID-TROSY experiments outperform the other two pulse sequences. We also compared the sensitivity of the experiments by plotting the curves of the average intensity divided by the square root of the total duration of a scan ($T_{scan}$), where $T_{scan}$ includes the duration of the interscan delay, the pulse sequence, and the acquisition time (Fig. 3B). As expected, the buildup curves increase at a short interscan delay, reaching a maximum value and then decreasing for longer interscan delays. Even in this representation, [¹H,¹⁵N] RAPID-TROSY performs similarly to the [¹H,¹⁵N] BEST-TROSY experiment for interscan delays less than 0.3 s and becomes the most sensitive experiment for interscan delays longer than 0.3 s. Figure 3C shows the sensitivity gain of the [¹H,¹⁵N] RAPID-TROSY relative to the other TROSY schemes for all the resolved amide resonances for D1 values of 0.2, 0.5, 1.0, and 2.0 s. At short interscan delays, we observed a significant sensitivity gain of the [¹H,¹⁵N] RAPID-TROSY over the [¹H,¹⁵N] TROSY-HSQC and [¹H,¹⁵N] WADE-TROSY experiments. However, the [¹H,¹⁵N] BEST-TROSY and [¹H,¹⁵N] RAPID-TROSY perform similarly, i.e., the distribution of the amide resonances for the [¹H,¹⁵N] BEST-TROSY is centered at 0% gain. At higher interscan delays, we observe a significant gain in sensitivity of the [¹H,¹⁵N] RAPID-TROSY relative to the other experiments, i.e., the sensitivity gain distributions are right-shifted. Figure 4 compares the [¹H,¹⁵N] RAPID-, BEST-, and TROSY-HSQC spectra acquired with interscan delays of 0.2 and 2.0 s. At an interscan delay of 0.2 s, the [¹H,¹⁵N] RAPID-TROSY and [¹H,¹⁵N] BEST-TROSY sequences perform similarly, with most peaks at the center of the bandwidth displaying similar intensity. However, a closer inspection reveals that several resonances

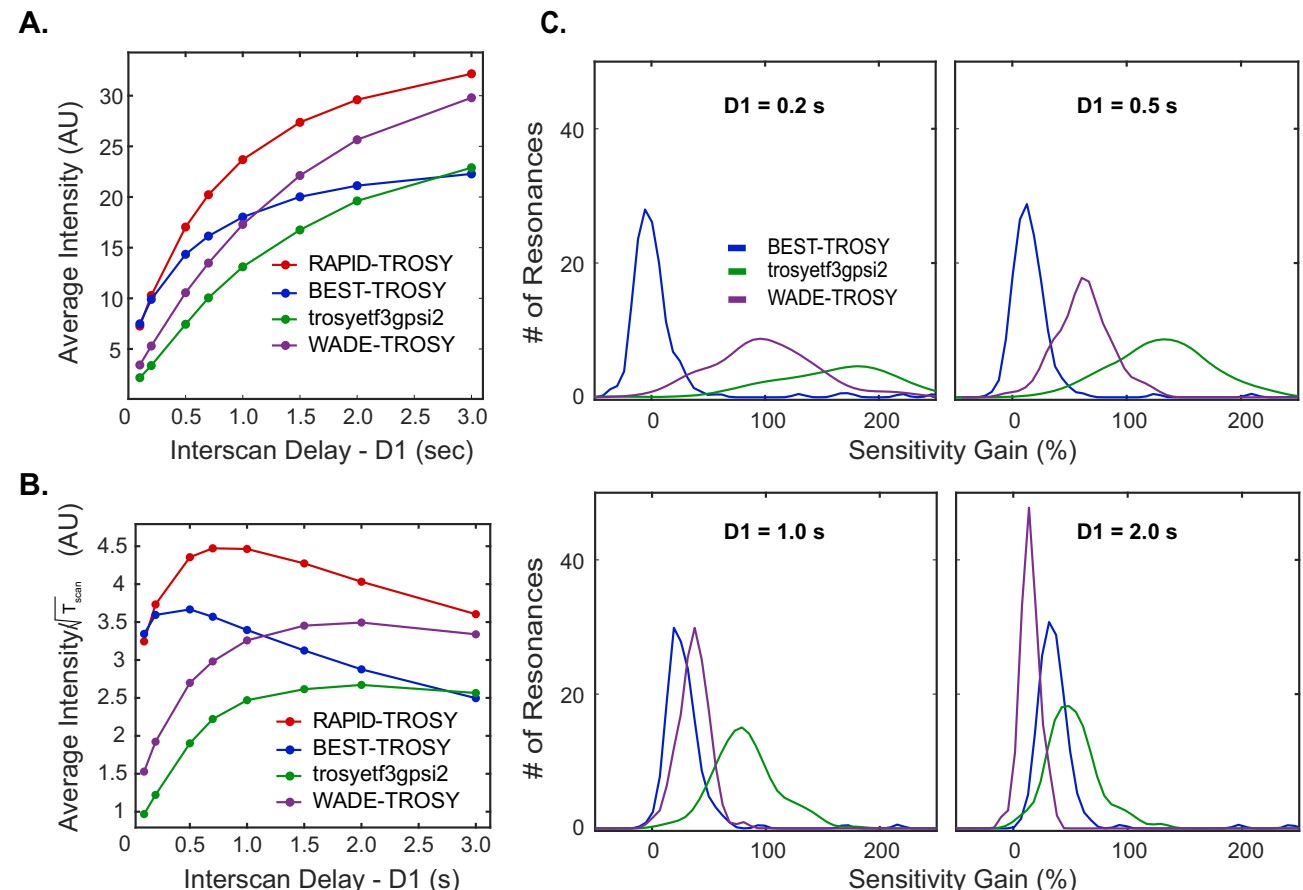

**Fig. 3 | Sensitivity comparison of the different TROSY schemes for RKIP.**
**A** Buildup of the average sensitivity calculated for 187 amide peaks of U-[15]N labeled
RKIP. **B** Sensitivity comparison of the different TROSY experiments as a function of
the interscan delay. The average intensity was divided by the root square of the
total scan duration, $T_{scan}$, which includes the duration of the pulse sequence, the
acquisition time, and the interscan delay. **C** Sensitivity gain distribution of RAPID-

TROSY for all the 187 resolved amide resonances of RKIP compared with the other
three pulse sequences at different interscan delays. The sensitivity gain for each
resonance is calculated using $(I_{RAPID} - I)/I$, where $I_{RAPID}$ indicates the intensity of
each RAPID-TROSY peak, and $I$ indicates the intensity of the corresponding reso-
nance in the other sequences.

in the [[1]H,[15]N] BEST-TROSY experiment at both ends of the [1]H band-
width are severely attenuated or undetectable (Fig. 4A, B), which is an
expected drawback of the band-selective pulses utilized for the BEST-
TROSY experiment[21]. The spectrum acquired with the [[1]H,[15]N] TROSY-
HSQC sequence (*trosyetf3gpsi2*) displays the lowest peak intensities
throughout the amide region, and although it shows signals at both
ends of the bandwidth, the suppression of the water signal for this
short interscan delay is inefficient (Fig. 4C). At an interscan delay of
2.0 s, the general features of the three experiments remain the same.
The longer interscan delays improve water suppression for the clas-
sical [[1]H,[15]N] TROSY-HSQC experiment. Although [[1]H,[15]N] BEST-TROSY
and the [[1]H,[15]N] TROSY-HSQC perform similarly for resonances at the
center of the bandwidth, the [[1]H,[15]N] RAPID-TROSY shows the highest
sensitivity across the entire operational bandwidth. The comparison of
the signal intensities among these experiments is highlighted in the 1D
traces of six selected resonances in Fig. 4D, E. Finally, the [[1]H,[15]N]
RAPID-TROSY experiment displays an improved suppression of the
side chain amide resonances due to the combination of phase cycling
and high-fidelity pulse operations.

To assess the performance of [[1]H,[15]N] RAPID-TROSY for rapidly
relaxing systems, we tested all the pulse sequences with two larger
proteins, U-[15]N labeled maltose binding protein (MBP, 42 kDa) and U-[15]N
labeled dimeric RIIβ subunit of PKA (RIIβ, 180 kDa; see Figs. 5 and 6, and
Supplementary Figs. 6 and 7). As for U-[15]N labeled RKIP, we tested
multiple D1 values. Again, the [[1]H,[15]N] TROSY-HSQC experiment
acquired with a D1 of 0.25 s displays the lowest average intensity for the

amide signals (Fig. 5B). In contrast, the [[1]H,[15]N] BEST-TROSY and [[1]H,[15]N]
RAPID-TROSY experiments show a significantly higher signal intensity.
As expected, the overall sensitivity of all experiments increases for
longer D1 values (Fig. 5B). However, the buildup of the average amide
intensity vs. D1 is different among the TROSY variants. For instance, the
[[1]H,[15]N] BEST-TROSY experiment increases up to a D1 value of 1.0 s and
reaches a plateau at equilibrium. In contrast, the average intensity for
the other TROSY experiments continues to increase to a D1 of 3s, with
the [[1]H,[15]N] RAPID-TROSY experiment reaching the highest average
intensity value. As a final test, we compared the performance of all
TROSY experiments for the dimeric U-[15]N RIIβ. The general trend of the
buildup curves for the average intensity *vs.* D1 is similar to that of MBP,
with the [[1]H,[15]N] RAPID-TROSY experiment displaying the highest
average sensitivity and outperforming all the other TROSY variants.
The slight variations in the buildup curves are likely due to the
relaxation behavior of the two proteins. Overall, the application of the
different TROSY schemes to these systems shows that the [[1]H,[15]N]
RAPID-TROSY experiment is the most sensitive over a wide range of
inter-scan delays and can be utilized either for regular or rapid pulsing
experiments.

## Discussion
NMR spectroscopy of biomacromolecules at high and ultra-high
magnetic fields presents significant technical challenges. First, the
broader chemical shift dispersion of the resonances requires higher RF
power to cover the entire bandwidth, a condition that often does not

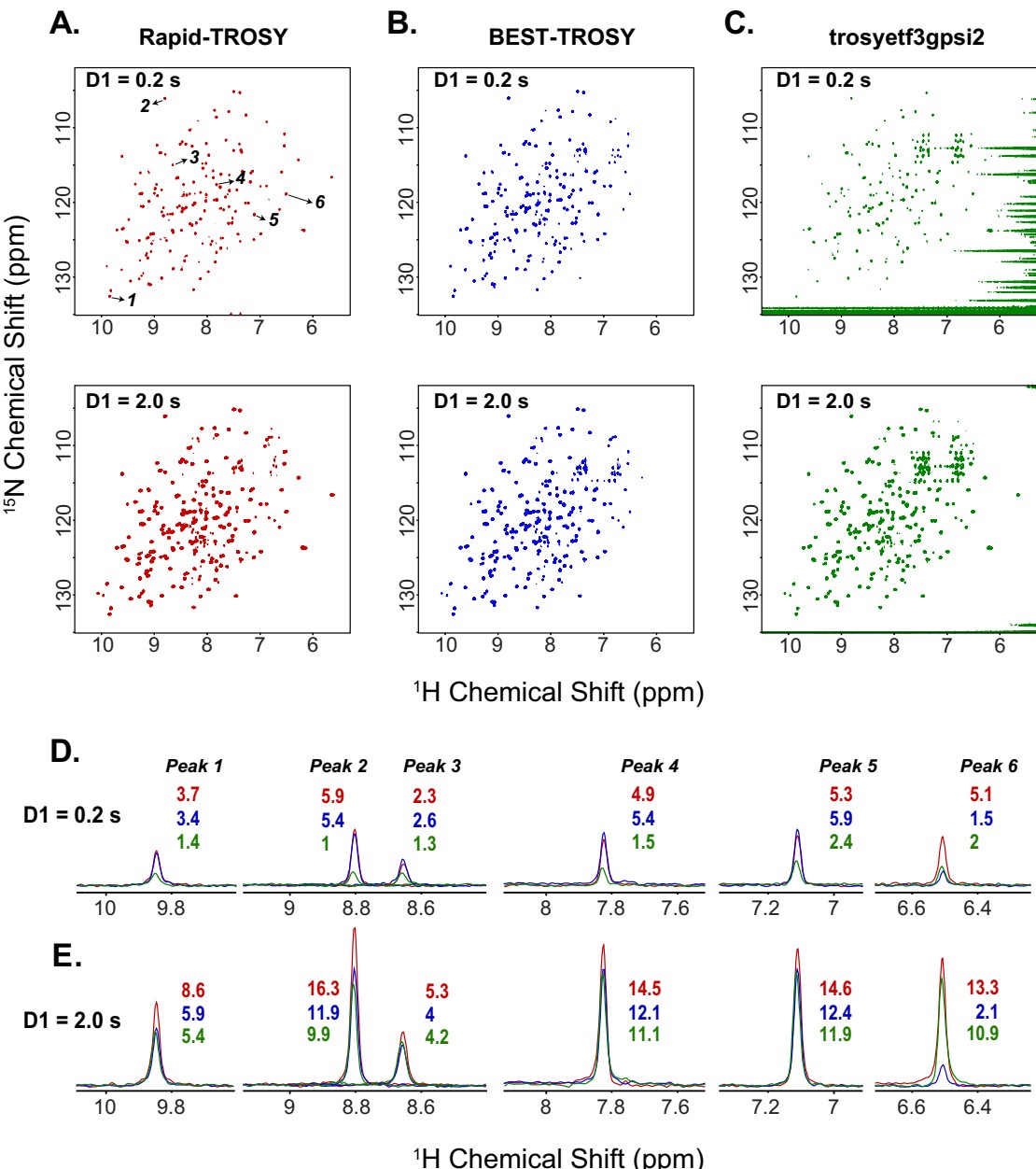

**Fig. 4 | Comparison of the performance of the different TROSY schemes with interscan delays of 0.2 and 2.0 s. A** RAPID-TROSY, **B** BEST-TROSY, and **C** conventional Bruker TROSY experiment (*trosyetf3gpsi2*). **D**, **E** 1D traces for the resonances 1–6 labeled in the **A** panel acquired with interscan delays of 0.2 and 2.0 s. The 1D spectra are analyzed at the same noise levels. The red, blue, and green 1D peaks correspond to the RAPID-TROSY, BEST-TROSY, and *trosyetf3gpsi2* experiments.

meet the specification of the probehead. Second, the enhancement of radiation damping makes water suppression problematic. Third, chemical exchange phenomena of labile sites, which are enhanced at high frequency, can significantly affect the intensity of the labile resonances (e.g., amide groups). Fourth, field inhomogeneity and pulse imperfections can significantly impact the sensitivity of the experiments even for simple spin operations[27,28]. Finally, the longitudinal relaxation times of the spin system demand longer interscan delays necessary to bring the nuclear spin magnetization to equilibrium, significantly extending the total experimental time.

A promising step forward was recently undertaken by Luy and coworkers[29], who used optimal control theory[30] to design band-selective 90° and 180° pulses that cover broader bandwidth and increase the sensitivity of triple-resonance experiments at 1.2 GHz[29], though this implementation is not suitable for fast pulsing techniques as yet. Here

we show that the updated version of GENETICS-AI can generate RF pulses for the excitation, inversion, and refocusing operations that address all the above issues. This software simultaneously searches for optimal amplitude and phase and generates RF pulse shapes with (a) high-fidelity of spin operations for minimizing RF artifacts, (b) broader operational bandwidths for carrying out spectroscopy at ultra-high magnetic fields, (c) WADE ¹H pulses for minimizing signal losses due to chemical exchange, (d) enable chemical shift and J coupling evolution during their execution, and (e) band-selective pulses on amide to enable fast pulsing (i.e., shorter inter-scan delays). The GENETICS-AI pulses are highly compensated for field inhomogeneity and irradiate broader bandwidths to cover the entire dispersion of chemical shifts for the current ultra-high field magnets (>1 GHz). The amplitude-phase modulated ¹H pulses do not irradiate the water signal; instead, they keep the bulk water magnetization along the z-axis, not affecting

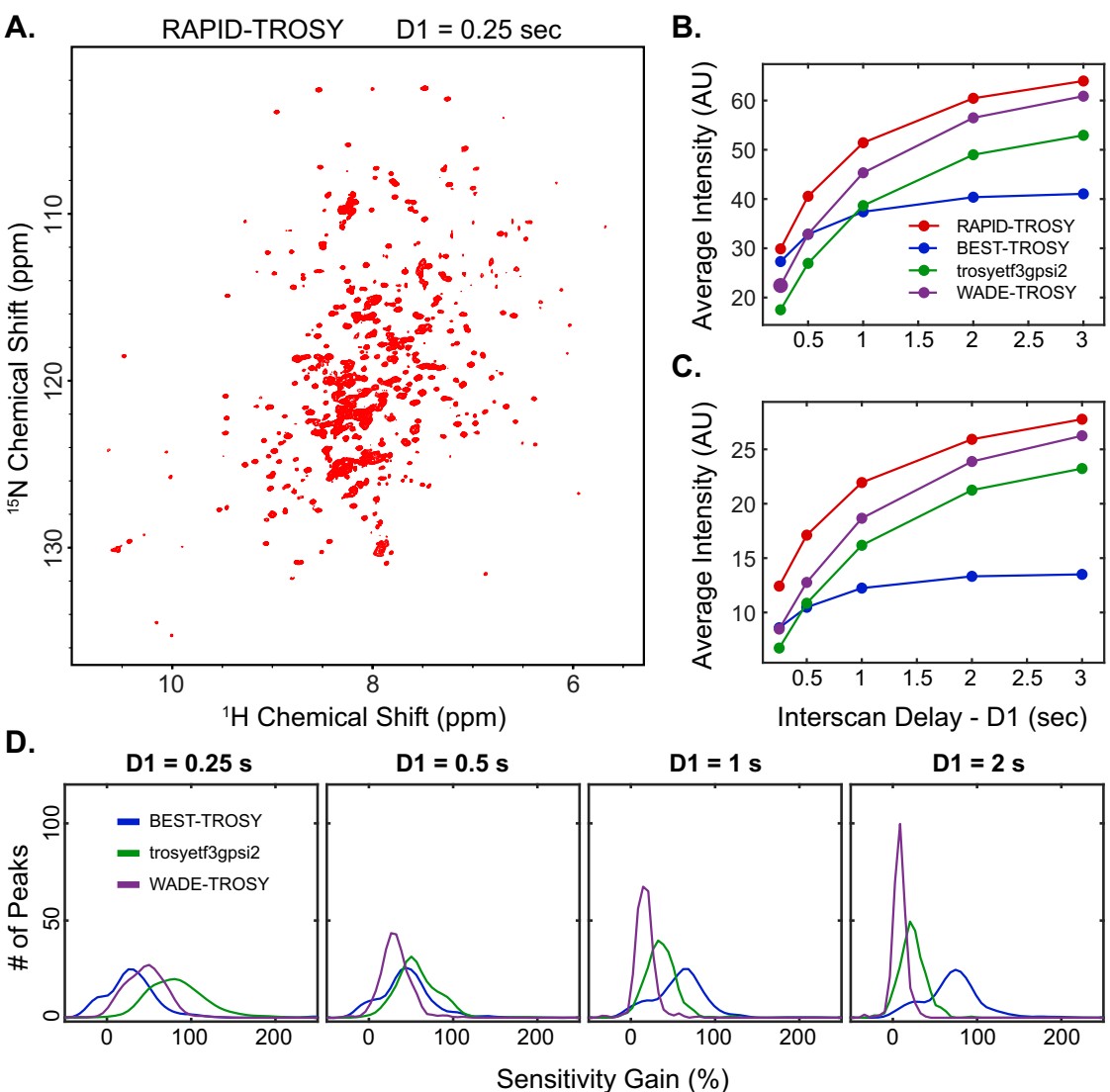

**Fig. 5 | Comparison of the different TROSY pulse sequences for MBP. A** RAPID-TROSY spectrum of U-$^{15}$N labeled MBP with an interscan delay of 0.25 s. **B** Average intensity of the 279 amide peaks vs. interscan delay for the four TROSY pulse sequences. **C** Average intensity of the 139 low-intensity (lower half) peaks vs. interscan delay. **D** Sensitivity gain distribution of RAPID-TROSY for all resolved amide resonances compared with the other three pulse sequences at different interscan delays. The sensitivity gain for each resonance is calculated using $(I_{RAPID} - I)/I$, where $I_{RAPID}$ indicates the intensity of each RAPID-TROSY peak, and $I$ indicates the intensity of the corresponding resonance in the other sequences. All spectra at different interscan delays are shown in Supplementary Fig. 6.

solvent-exchanging sites. Another significant advantage of our RF pulses is the concurrent evolution of chemical shift and J coupling during the pulse's execution. Therefore, replacing all RF pulses in a multipulse sequence with equivalent pulse operations with embedded chemical shift and J coupling evolution will significantly reduce the overall duration of the pulse sequence, minimizing signal losses due to nuclear spin relaxation[31]. While this aspect has a negligible impact on small-size proteins, it will influence the design and efficiency of triple resonance experiments for large macromolecules, where transverse relaxation is sizable. Also, the amplitude-phase modulated $^{15}$N pulses guarantee high fidelity for any spin operations, avoiding typical artifacts observed for classical square (hard) pulses. More importantly, these pulses can be implemented in fast pulsing experiments, reducing the overall experimental time with a gain in terms of time and costs. Although these RF pulses could be implemented into the triple-resonance sequences, the flexibility of the GENETICS-AI software makes it possible to tailor the RF shapes based on the biomolecule's chemical shift fingerprint or spectrometer characteristics. For example, it is possible to change water selectivity by reducing or increasing

the RF amplitude of the UAev1 and UARev1 pulses or design specific pulse operations with higher compensation for RF, probes, or magnet RF inhomogeneities.

In conclusion, the combination of an evolutionary algorithm and AI made it possible to design high-fidelity and highly compensated RF pulse shapes with variable amplitude and phase that allow J coupling and chemical shifts to evolve during their execution. When implemented in a TROSY sequence, these RF shapes shorten the total experimental time without compromising the irradiation bandwidth and water suppression, minimizing solvent-exchange phenomena. The RAPID-TROSY experiment will find immediate applications in acquiring spectra for large, poorly soluble proteins at very low concentrations, unstable samples, biomolecules for which isotopic enrichment is not possible, nucleic acid analysis[32], in-cell NMR spectroscopy[33,34], metabolomics, real-time biochemical processes such as kinetics of aggregation and oligomerization[35,36], light-activated conformational transitions[37], as well as drug/ligand screening[38]. Finally, the RAPID-TROSY experiment developed here constitutes a template for designing a new class of double- and triple-resonance fast pulsing

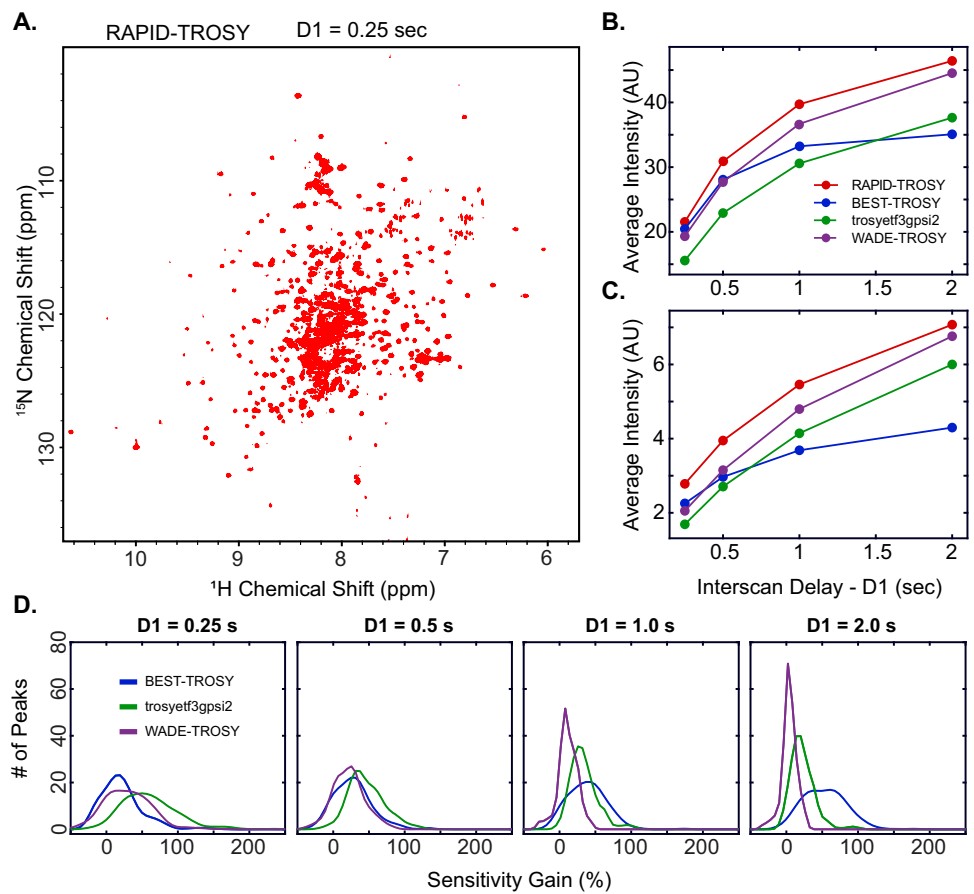

**Fig. 6 | Comparison of the different TROSY pulse sequences for RIIβ dimer.**
**A** RAPID-TROSY of $^1$H,$^{15}$N labeled PKA RIIβ dimer acquired with an interscan delay of 0.25 s. **B** Average intensity of the amide peaks vs. interscan delay for all resolved peaks (244); **C** Average intensity of the less intense peaks (122, lower half) vs. interscan delay. **D** Sensitivity gain distribution of RAPID-TROSY for all the resolved 390 amide resonances of RIIβ compared with the other three pulse sequences at different interscan delays. All spectra at different relaxation delays are shown in Supplementary Fig. 7.

sequences to accelerate the characterization of large biomacromolecules at high and ultra-high magnetic fields.

## Methods

### Expression and purification of RKIP

Recombinant $^{15}$N labeled RKIP was expressed and purified, as reported previously[39]. Briefly, *E. coli* BL21(DE3) pLysS cells (Invitrogen™, ThermoFisher Scientific) were transformed with a pE-SUMO vector (ThermoFisher Scientific) encoding for the rat gene of RKIP (*Rattus norvegicus*, PEBP1). After an overnight proliferation in LB media, the cells were transferred to an M9 minimal media containing $^{15}$NH$_4$Cl (Cambridge Isotope Laboratories Inc. - CIL). Protein overexpression was initiated with 0.4 mM Isopropyl β-ᴅ-1-thiogalactopyranoside (IPTG) when the cells reached an optical density at 600 nm (OD$_{600}$) of 1.1 and it was carried out for 5 hours at 30 °C. The cells were harvested by centrifugation at 6370 g for 30 min, and the temperature was held at 4 °C. The cell pellet was collected in a conical tube and stored at −20 °C. For protein purification, the cell pellet was resuspended in 50 mM Tris-HCl buffer (pH 8.0) containing 100 mM NaCl, 20% sucrose, 0.15 mg/ml lysozyme, one tablet of protease inhibitor (cOmplete™, Roche Applied Science), 100 U/mL DNAse I (Roche Applied Science), and 5 mM β-mercaptoethanol (β-me). The cells were lysed using a French press at 1000 psi. Cell debris was centrifuged at 45,700 g for 40 min (4 °C), and the supernatant containing the target protein was mixed with Ni$^{2+}$-NTA agarose affinity resin (ThermoFisher Scientific). The resin was loaded into a column and washed using 50 mM Tris-HCl buffer (pH 8.0) with 100 mM NaCl, 20% sucrose, 1 mM

phenylmethylsulfonyl fluoride (PMSF), and 5 mM β-me. RKIP was eluted in two fractions using the same 200 mM and 500 mM imidazole buffer. The elution fractions were combined and transferred into a dialysis bag with a stoichiometric amount of recombinant UPL1 protease and dialyzed against 50 mM Tris-HCl buffer (pH 8.0), 150 mM NaCl, 5 mM β-me, 0.5 mM PMSF at 4 °C. A reverse Ni$^{2+}$-NTA purification was performed to eliminate other protein contaminants. The fractions containing un-tagged RKIP (flow-through) were concentrated using a 10 kDa spin concentrator (Sigma-Aldrich, Merck) and loaded into a 16/60 Hi Load Superdex 200 (GE Healthcare) size exclusion column (SEC) using 50 mM Tris-HCl buffer (pH 8.0), 150 mM NaCl, 2 mM dithiothreitol (DTT) as a mobile phase. The purified protein was concentrated and stored at 4 °C in the SEC buffer with protease inhibitors. $^{15}$N labeled RKIP sample was prepared by buffer-exchanging and concentrating the protein in 20 mM KH$_2$PO$_4$ (pH 6.5), 10 mM DTT, 10 mM MgCl$_2$, and 1 mM NaN$_3$ buffer supplied with 5% D$_2$O, and 0.5% Pefa block® (Sigma-Aldrich, Merck) to a final protein concentration of 0.3 mM.

### Expression and purification of MBP

Recombinant U-$^{13}$C,$^{15}$N labeled MBP was expressed and purified from E. coli bacteria[22]. Briefly, competent NEB Express cells (New England Biolabs - NEB), transformed using the pMAL-c6T vector (NEB), were cultured in M9 minimal medium containing $^{15}$NH$_4$Cl (CIL) and $^{13}$C-glucose (CIL) as the only nitrogen and carbon sources, respectively. Protein expression was induced using 1 mM of IPTG at OD$_{600}$ 0.8. The cells were cultured at 30 °C and harvested after 5 hours by

centrifugation for 30 min at 6370 g and 4 °C. The cell pellet was resuspended in 20 mM phosphate buffer (PBS, pH 7.3), 120 mM NaCl, 1 mM ethylenediaminetetraacetic acid (EDTA), 0.15 mg/mL lysozyme, 0.05% glycerol, 2 mM DTT, 1 tablet of protease inhibitor (cOmplete™, Roche Applied Science), and 100 U/mL DNAse I (Roche Applied Science) and was homogenized using a cell grinder. The resuspension was then sonicated using Branson Sonifier 450 (output of 4; duty cycle, 40%) for 10 min. Cell debris was pelleted by centrifugation at 45,700×$g$ for 20 min at 4 °C, and the pooled supernatant containing MBP was incubated with amylose resin (New England Biolabs, USA). The suspension was added to a gravity column and washed with 20 mM PBS (pH 7.3), 120 mM NaCl, and 1 mM EDTA buffer. Next, MBP was eluted using ~100 mL of 20 mM PBS (pH 7.3), 120 mM NaCl, and 1 mM EDTA buffer containing 60 mM maltose. A final purification step was obtained with the size exclusion chromatography (16/60 Hi Load Superdex 200, GE Healthcare Life Sciences), using 10 mM Na$_2$HPO$_4$, 0.1 mM EDTA, and 1 mM NaN$_3$ buffer as a mobile phase. The purified MBP was then collected and stored at 4 °C. The NMR sample was prepared by concentrating the stored protein to a final concentration of 1 mM using a 10 kDa spin concentrator (Sigma-Aldrich, Merck). The final sample contained 5% D$_2$O and 0.5% Pefa block® (Sigma-Aldrich, Merck).

## Expression and purification of RIIβ dimer

The expression vector encoding for the bovine RIIβ subunit of cAMP-dependent protein kinase A (PKA) was a gift from Dr. Susan Taylor (University of California at San Diego, La Jolla, CA, USA). The protein was expressed in *E. coli* BL21 (DE3) pLysS cells (Invitrogen, Thermo-Fisher Scientific) in M9 media supplemented with $^{15}$NH$_4$Cl. Protein overexpression was induced with 0.4 mM IPTG and carried out for 11 hours at 20 °C. The RIIβ purification procedure was adapted from the murine catalytic subunit (Cα) protocol[40]. Briefly, RIIβ purification was carried out by co-immobilized metal affinity chromatography (IMAC) with human catalytic subunit (hPKA-Cα), followed by a denaturation/renaturation step to remove the cyclic adenosine monophosphate (cAMP), and a final size exclusion step. The IMAC was carried out using a Ni$^{2+}$-NTA agarose affinity resin (ThermoFisher Scientific), combining RIIβ and hPKA-Cα lysate. Note, the details of this step are reported in the protocols described by Walker et al.[41]. and Olivieri et al.[42]. RIIβ was eluted using 30 mM MOPS (pH 8.0), 15 mM MgCl$_2$, 25 mM KCl, 0.1 mM PMSF, and 5 mM β-me supplemented with 500 μM cAMP. The eluted RIIβ was unfolded and refolded using the protocol described by Zhang et al.[43]. Following the refolding step, the protein was loaded into a 16/60 Hi Load Superdex 200 (GE Healthcare) gel filtration column using 50 mM Tris-HCl buffer (pH 7.5) as mobile phase with 200 mM NaCl, 2 mM EDTA, 2 mM ethylene glycol- bis(β-aminoethyl)-N,N,N′,N′-tetra acetic acid (EGTA), and 5 mM DTT. The purified protein was stored in the gel filtration buffer supplemented with 30% glycerol at −20 °C. For the NMR experiment, the appropriate amount of protein was dialyzed against 2 L of 20 mM KH$_2$PO$_4$ (pH 6.5), 10 mM DTT, 10 mM MgCl$_2$, and 1 mM NaN$_3$ buffer. The protein was then concentrated to 0.25 mM, supplied with 5% D$_2$O and 0.5% Pefa block® (Sigma-Aldrich, Merck), and loaded into a 5 mm Shigemi® NMR tube.

## NMR Spectroscopy

NMR experiments were conducted on a 900 MHz Bruker Avance III spectrometer equipped with a TCI cryoprobe and operated by Bruker TopSpin v3.6.1. The temperature was held constant at 300 K. The intensity buildup curves were obtained varying the inter-scan delay from 0.2 s to 3 s. All 2D TROSY-HSQC experiments were acquired with 1024 complex points in the direct dimension and 96 increments in the indirect dimension for RKIP, 160 for MBP, and 128 for the RIIβ subunit dimer. The total number of scans was 16 for RKIP, 4 for MBP, and 32 for RIIβ subunit dimer. The indirect

dimensions were acquired with a States-TPPI phase incrementation. All spectra were processed using NMRPipe Version 11.4[44] using a sine-bell apodization function with an offset of 0.4 in both dimensions. Before Fourier transform, the complex matrices were zero-filled to a final size of 256 × 1536 points. All NMR spectra were analyzed by NMRFAM-SPARKY v1.470[45].

## Computer simulations

All simulations and pulse design were conducted using Matlab® (R2022a) on a desktop computer equipped with an intel core i7 processor. The offset responses for all pulses were simulated using a single spin system. The initial states of the magnetization evolved under the effects of both RF pulse and chemical shift Hamiltonian, with an offset ranging from −10 to +10 kHz for the latter. The offset responses were generated by calculating the x, y, and z components of the magnetization (M$_x$, M$_y$, and M$_z$) in the final state. A two-spin system was simulated with a scalar coupling of −92 Hz to determine the J coupling evolution. Starting with a known single spin state, we calculated the two-spin terms created in the final state and quantified the scalar coupling evolution during the pulse. Finally, relaxation responses were simulated using the single-spin Bloch equation. The longitudinal ($R_1$) and transverse ($R_2$) relaxation times varied from 0 to 50 s$^{-1}$, and the magnitude of the final state was calculated for each pulse.

## Reporting summary

Further information on research design is available in the Nature Portfolio Reporting Summary linked to this article.

## Data availability

All source data are available at the University of Minnesota Repository Site (https://hdl.handle.net/11299/254765). Source data are provided with this paper.

## Code availability

The AI-designed RF pulses and the [$^1$H,$^{15}$N] RAPID-TROSY HSQC pulse sequence are available at the University of Minnesota Repository Site (https://hdl.handle.net/11299/254765) and GitHub (https://github.com/manuvsub/RAPID-TROSY).

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

## Acknowledgements

The authors acknowledge Prof. M. Latham (UMN) for carefully reading the manuscript. All NMR experiments were conducted at the Minnesota NMR Center (https://nmr.umn.edu/). This work was supported by the National Institute of Health (HL144130 to G.V.) and a subcontract to G.V. from GM121735 (Marsha Rosner, P.I.).

## Author contributions
M.V.S. designed the pulses, implemented the pulse sequences, and performed the NMR experiments. C.O. prepared the protein samples and contributed to the NMR data analysis. G.V. designed the research and analyzed the NMR data. M.V.S., C.O., and G.V. wrote the paper. All the authors have seen and approved the submitted manuscript.

## Competing interests
G.V. and M.V.S. are inventors of the GENETICS-AI software, patent US 11,221,384. G.V. is the founder of Kantika LLC. C.O. declares no competing interests.
