## [Peer review file · Nature Communications]

REVIEWERS' COMMENTS

Reviewer #1 (Remarks to the Author):

This is a very good paper that makes use of AI to optimize the shapes (waveforms) of amide-selective 1H and 15N pulses to avoid excitation of water and aliphatic resonances. It can be accepted with very minor revision. Not entirely sure why this was submitted to Nature Comm rather than JACS or Nature Methods or Nature Chemistry, but that is the authors' choice.

One comment that the authors should perhaps discuss is that I really don't know whether this will find widespread use as a 2D 1H-15N TROSY is the shortest (simplest) experiment that most people use to test samples, and as such they don't really care whether the acquisition time is say 30 min or 2 hrs. Now obviously when the authors' approach is implemented for 3D and 4D experiments, that's a whole other matter, but once one demonstrates the implementation for the 2D, extension to 3D and 4D is conceptually trivial. Perhaps the authors could mention examples of cases where rapid recording of a TROSY experiment is really useful. e.g. to follow the time course of aggregation processes immediately comes to mind.

Minor point:

1. The WADE-TROSY experiment, which in many plots gives comparable results to the RAPID-TROSY (the experiment describe din the current paper), is not described and not published yet (ref. 22 in press). Personally I don't know what the WADE-TROSY actually is (and hadn't heard of it before), and if the paper isn't yet out obviously I cannot know. Therefore, I think a brief description would be helpful to the reader.

2. In ref. 8, one of the authors, Bodenhausen, who incidentally is the originator of the standard HSQC experiment, has been accidentally left out.

Reviewer #2 (Remarks to the Author):

Accelerating biomolecular NMR spectroscopy... describes a series of AI generated pulses that have either excellent selectivity (over a desired bandwidth) or that are broadbanded to achieve uniform excitation. The pulses allow for control of water magnetization and are highly compensated for inhomogeneity. Importantly they also allow for evolution of magnetization due to scalar coupling in the case of 180o pulses where one wants this to occur. With the development of new instrumentation pushing to higher magnetic fields the importance of optimized pulses becomes clear. I believe that this work is an excellent step in this direction. I have only a few questions.

- 1) I would appreciate some discussion about how these long pulses affect the phase of the resulting magnetization (for example after the UA90ev1 pulse what is the phase of magnetization – does a UA90ev1 pulse along X take magnetization from Z to -Y?
- 2) On line 133 the authors state that the UA90ev1 pulse leave magnetization between 0 and -7.5 kHz unaffected? Yet it seems from Fig. 1A that at -6 ppm and further upfield there is some effect.
- 3) When the authors are discussing Figure 1, I don't see anything relating to 1D.
- 4) Is there ever a case where one must worry about time-reversed pulses?
- 5) In your pulse sequences, water magnetization is along -Z during t1. It is known that it will recover to Z due to radiation damping and I imagine that this could happen during t1. Why not use bipolar gradients during t1?
- 6) Finally, in studies of high molecular weight proteins, deuteration is often applied, mitigating the improvements associated with using the large (now not) reservoir of proton spins that are unperturbed. What are the gains now in comparison to other pulse sequences? That is focusing just on the improved inhomogeneity effects are the gains substantial?

Reviewer #1

AU: We thank this reviewer for carefully reading the manuscript, for positive feedback, and for suggestions that helped improve the quality and importance of this new method. Below are our responses to the specific critiques and queries.

R#1: One comment that the authors should perhaps discuss is that I really don't know whether this will find widespread use as a 2D 1H-15N TROSY is the shortest (simplest) experiment that most people use to test samples, and as such they don't really care whether the acquisition time is say 30 min or 2 hrs. Now obviously when the authors' approach is implemented for 3D and 4D experiments, that's a whole other matter, but once one demonstrates the implementation for the 2D, extension to 3D and 4D is conceptually trivial. Perhaps the authors could mention examples of cases where rapid recording of a TROSY experiment is really useful. e.g. to follow the time course of aggregation processes immediately comes to mind.

AU: We agree with these comments. Of course, the 2D RAPID-TROSY scheme is only the first step for the implementation of AI-designed pulses into 2D and pseudo-3D experiments for nuclear spin relaxation experiments as well as 3D and 4D pulse sequences to speed up sequential resonance assignments and structure determination of biomacromolecules. Indeed, we agree that this basic experiment can be utilized for kinetic experiments, following the time course of relatively slow processes such as biomolecular aggregation and oligomerization. Other applications include metabolomics, in-cell NMR, macromolecules with low solubility, and drug screening. Following this reviewer's suggestions, we included a few sentences in the discussion of the manuscript.

Minor point:

1. The WADE-TROSY experiment, which in many plots gives comparable results to the RAPID-TROSY (the experiment described in the current paper), is not described and not published yet (ref. 22 in press). Personally, I don't know what the WADE-TROSY actually is (and hadn't heard of it before), and if the paper isn't yet out obviously, I cannot know. Therefore, I think a brief description would be helpful to the reader.

AU: At the time of submission, the paper cited in ref. 22 was in press. We now provide a complete reference for this paper. However, the appropriate reference is now 23 in which we report the WADE pulses. Paraphrasing our PCCP article, the WADE-TROSY utilizes Water Irradiation DEvoided (WADE) refocusing pulses in the 1H channel for the suppression of the WATER signal. These WADE pulses are designed with ' π -shifted' symmetry, ensuring a null point precisely at the water resonance. Compared to traditional water suppression methods, WADE pulses exhibit greater sensitivity and offer adjustable water selectivity to prevent the suppression of 1H resonances near the water signal. We included a few sentences in this manuscript explaining the original design of the WADE pulses.

2. In ref. 8, one of the authors, Bodenhausen, who incidentally is the originator of the standard HSQC experiment, has been accidentally left out.

AU: We apologize for this mistake. Indeed, Dr. Bodenhausen pioneered this method. We have corrected our citation library and the manuscript.

Reviewer #2 (Remarks to the Author):

We appreciate this reviewer's positive feedback, pointing out the importance and impact that this new method will have in the NMR and structural biology community. The reviewer's comments/critiques are addressed below.

1) I would appreciate some discussion about how these long pulses affect the phase of the resulting magnetization (for example after the UA90ev1 pulse what is the phase of magnetization – does a UA90ev1 pulse along X take magnetization from Z to –Y?)

AU: We agree our statements in the paper need further clarification. The pulses developed in this paper do not affect the phase of the magnetization. For instance, the UA90ev1 pulse was evolved using GENETICS-AI to include both the single pulse operation and two delays, i.e., these pulses contain two operations. To clarify this point, we expanded the discussion in the main text. Below is a brief description of each pulse to clarify this point.

The UA90ev1 pulse excites selectively the amide protons and includes delays to allow for chemical shift and J coupling evolution. Specifically, the UA90ev1 pulse includes a 90° pulse sandwiched between delays of half the pulse length. The 90° operation acts as a universal flipping operation, converting the z magnetization into -y. The subsequent delay after the 90° pulse causes the dephasing of the magnetization, whose extent depends on the pulse length and chemical shift (Figure 1A). However, the pulse and delays are part of the entire spin echo sequence, with no net effects on the magnetization evolution and the relative phase.

The UARev1 and UARev2 pulses perform refocusing operations. When applied along the x-direction, these pulses invert the y and x magnetization (Figure 1B and 1C). These pulses include a 180° pulse sandwiched between two delays that are 0.49 times the length of each pulse. Since the 180° operation is centered relative to these delays, the dephasing of the magnetization caused by the first delay is refocused during the second delay. When applied simultaneously with a 180° pulse on the second channel, these delays cause the magnetization to evolve under the J coupling. This evolution can be exploited during spin echo sequences for polarization transfer, reducing the length of the spin echo. In both cases, however, there is no evolution of the phase and no phase correction.

Finally, the UARev1 pulse refocuses the amide magnetization, avoiding the water signal. The pulse includes a delay of 0.75 times the total pulse duration. Figure 1D shows the response of the magnetization to this pulse for initial states M_x , M_y , and M_z .

2) On line 133 the authors state that the UA90ev1 pulse leave magnetization between 0 and -7.5 kHz unaffected? Yet it seems from Fig. 1A that at -6 ppm and further upfield there is some effect.

AU: We thank this reviewer for pointing this out. This is a typo. We corrected the main text: “[...] the UA90ev1 pulse leaves the magnetization between 0 and -6.2 kHz unaffected.”

3) When the authors are discussing Figure 1, I don't see anything relating to 1D.

AU: We now include the description of panel 1D in the main text.

4) Is there ever a case where one must worry about time-reversed pulses?

AU: For the RAPID-TROSY sequence, the evolutions during all the pulses are either utilized as a part of the spin echo or compensated by 180 pulses. Therefore, there is no need to worry about the time-reversed pulses.

5) In your pulse sequences, water magnetization is along -Z during t1. It is known that it will recover to Z due to radiation damping and I imagine that this could happen during t1. Why not use bipolar gradients during t1?

AU: We thank this reviewer for pointing this out. The original scheme of the RAPID-TROSY available on GitHub already incorporates the bipolar gradients. We accidentally omitted from Figure 2. We have now corrected this figure including the bipolar gradients.

6) Finally, in studies of high molecular weight proteins, deuteration is often applied, mitigating the improvements associated with using the large (now not) reservoir of proton spins that are unperturbed. What are the gains now in comparison to other pulse sequences? That is focusing just on the improved inhomogeneity effects are the gains substantial?

AU: For fully perdeuterated proteins, the improvement of the RAPID-TROSY results from the inhomogeneity compensation, higher fidelity, broader bandwidth, and shorter pulse sequence. These improvements are still significant, especially for biomacromolecules with broad chemical shift dispersion and NMR experiments at ultra-high fields (> 900 MHz). Based on our initial tests, these improvements will impact mostly 3D and 4D experiments. Indeed, we are in the process of studying the effects of partial deuteration and relaxation enhancement agents to make RAPID-TROSY more efficient for larger macromolecules. These efforts will be part of our future endeavors.